# Gender-Specific Determinants of eHealth Literacy: Results from an Adolescent Internet Behavior Survey in Taiwan

**DOI:** 10.3390/ijerph19020664

**Published:** 2022-01-07

**Authors:** Chia-Shiang Cheng, Yi-Jen Huang, Chien-An Sun, Chi An, Yu-Tien Chang, Chi-Ming Chu, Chi-Wen Chang

**Affiliations:** 1Graduate Institute of Life Sciences, National Defense Medical Center, Taipei 114201, Taiwan; bear654328@gmail.com; 2School of Public Health, National Defense Medical Center, Taipei 114201, Taiwan; yijen321@gmail.com; 3Department of Public Health, College of Medicine, Fu-Jen Catholic University, New Taipei City 242062, Taiwan; 040866@mail.fju.edu.tw; 4Big Data Research Center, Fu-Jen Catholic University, New Taipei City 242062, Taiwan; 5School of Nursing, College of Medicine, Chang Gung University, Taoyuan City 33302, Taiwan; anchi3910@gmail.com (C.A.); cwchang@mail.cgu.edu.tw (C.-W.C.); 6Department of Psychiatry, Lin Kou Chang Gung Memorial Hospital, Taoyuan City 33302, Taiwan; 7Division of Biostatistics and Medical Informatics, Department of Epidemiology, School of Public Health, National Defense Medical Center, Taipei 114201, Taiwan; 8Graduate Institute of Medical Sciences, National Defense Medical Center, Taipei 114201, Taiwan; 9Department of Public Health, School of Public Health, China Medical University, Taichung 404328, Taiwan; 10Department of Public Health, Kaohsiung Medical University, Kaohsiung 807378, Taiwan; 11Division of Pediatric Endocrinology & Genetics, Department of Pediatrics, Chang-Gung Memorial Hospital, Taoyuan City 33302, Taiwan

**Keywords:** gender health information, adolescent, internet, internet rumors, eHealth Literacy

## Abstract

Adolescents’ Internet health information usage has rarely been investigated. Adolescents seek all kinds of information from the Internet, including health information, which affects their Health Literacy that eHealth Literacy (eHL). This study is a retrospective observational study, we have total of 500 questionnaires were distributed, 87% of which were recovered, and we explored the channels that adolescents use to search for health information, their ability to identify false information, and factors affecting the type and content of health information queried. Adolescents believe that the Internet is a good means to seek health information because of its instant accessibility, frequent updating, convenience, and lack of time limits. More boys use the Internet to seek health information than girls in junior high schools (*p* = 0.009). The Internet is an important source of health information for adolescents but contains extensive misinformation that adolescents cannot identify. Additionally, adolescent boys and girls are interested in different health issues. Therefore, the government should implement measures to minimize misinformation on the Internet and create a healthy, educational online environment to promote Adolescents’ eHealth Literacy (eHL).

## 1. Introduction

Internet use has rapidly become a common activity for young people [1], and adolescents have integrated Internet use into many aspects of their daily lives [2]. Because of its characteristics of anonymity and privacy, the Internet has become an important source of knowledge for young people, who use it as a tool for education and access to health information [3].

According to a survey by the National Center for Health Statistics (HCHS), 61% of American adults (18–64 years) search for health information online, but more than 70% of the information obtained is incorrect [4]. Parents are often adolescents’ main source of health information, but because of poor health literacy among both parents [5] and adolescents, young people mistakenly believe Internet misinformation about health. Therefore, increased attention to adolescents’ Internet health information acquisition environment is important [6,7].

“The ability to seek, find, understand, and appraise health information from electronic sources and apply the knowledge gained to addressing or solving a health problem”, Norman and Skinner defined eHealth Literacy in 2006 [8]. Because eHealth Literacy and Media Health Literacy are critical for enabling individuals to actively participate in their own health, there is a growing body of work examining the potential and efficacy of treatments to improve these abilities [9].

According to a study, 52–80 percent of Internet users look for health information on the internet. People are aware that health information found on the Internet is not always accurate, but there is no way to properly determine if the material is accurate. The research only looks for adult-related research [10]. Another research evaluated the eHealth Literacy of two ethnic groups: traditional college students (18–22 years old) and older adult students (55–72 years old). The study’s findings suggest that conventional college students frequently utilize websites to solve health concerns, whereas older adult students communicate health information using communication software [11].

The capacity to read and understand electronic health information is critical, especially in light of the COVID-19 epidemic. According to a poll of 1074 persons in the United States, the majority of adults have inadequate eHealth Literacy when it comes to coronavirus. The author thinks that the quality of COVID-19 information on the Internet should be increased, or that health information search abilities should be enhanced [12]. Another eHealth Literacy survey in the United States found that just 49.1 % of Internet users (aged 55 and over) had strong eHealth Literacy. The study also advised that education for older individuals (55 and up) should be enhanced. Literacy in eHealth [13].

However, adolescent eHealth Literacy has received little attention [14], and more research is critically needed. The means by which adolescents seek online health information, as well as their capacity to recognize incorrect information and the factors that influence the type and content of health information requested, were investigated in this study.

## 2. Materials and Methods

### 2.1. Study Population

Students in two classes at a public junior high school and high school in Taipei, Taiwan, were randomly selected in a stratified manner based on student status and grade for participation in a cross-sectional questionnaire survey. A total of 500 copies of the questionnaire were issued, 437 of which were recovered, yielding a recovery rate of 87%. Among the respondents, female students constituted the majority (57%), and the age range was 13–20 years (Table 1).

### 2.2. Questionnaire

The questionnaire contained three parts: (1) basic information; (2) Internet usage; and (3) Internet health and hygiene information usage. The content of the questionnaire was reviewed by two experts in the field of health information (expert validity) who provided revision recommendations after reviewing the completed questionnaires of five junior high school and high school students. The ethical approval from the Tri-Service General Hospital’s Institutional Review Board, with the number “TSGH-1-105-05-081”.

### 2.3. Importance-Performance Analysis

The importance-performance analysis (IPA) method was adopted to analyze the adolescents’ assessment of the importance of and satisfaction with health information channels. In this method, the *Y*-axis represents satisfaction, the *X*-axis represents importance, and four quadrants are generated based on importance and satisfaction. Of these, Quadrant I is the area with high importance and high satisfaction, Quadrant II is the area with low importance and high satisfaction, Quadrant III is the area with low importance and low satisfaction, and Quadrant IV is the area with high importance and low satisfaction, representing an area requiring active improvement.

### 2.4. Statistical Analysis

Descriptive statistics were used to analyze differences in the distribution of demographic variables, Internet usage, Internet search channels, and health information queries between junior high school and high school students and between male and female students. Differences between classes and genders were tested using Student’s *t*-test or the chi-square test, and significance was set at *p* < 0.05. Statistical analyses were performed using the Statistical analysis software SPSS 18.

## 3. Results

### 3.1. Internet Use: First Use, Frequency, and Purpose

More than 95% of the adolescents started accessing the Internet before or in elementary school, more than 80% of the students spent more than one hour on the Internet per online session on average in the past month, and approximately 90% of the students used the Internet at least once in a week on average (Table 2). Most of the students used the Internet at home (95.9%) (Table 3), mostly for entertainment(Table 4), and most often searched for video and audio content(Table 5). The frequency at which the students browsed health-related news or information per week was low, and 70% of the adolescents browsed health-related news or information less than once per week (Table 6). The purpose of Internet use and the content browsed differed by age and gender (Table 4 and Table 5).

### 3.2. Internet Usage Frequency and Advantages by Gender

The students’ main sources of health-related information included television news, the Internet, and teachers (Table 7). More boys used the Internet to seek health information than girls in junior high schools (*p* = 0.009). The health content searched by junior high school girls included “weight loss” (p < 0.001) and “beauty and health” (*p* < 0.001). The health content searched by high school girls included “mental health” (*p* = 0.004), “weight loss” (*p* < 0.001), and “beauty and health” (*p* < 0.001). (*p* < 0.001). The health content searched by high school boys included “sex knowledge” (*p* < 0.001). More than 90% of the students used the Internet to seek health information because the Internet provides instant and convenient access to the latest information without time limits. More than 90% of the students wanted health information sites to have a search function with clearly defined categories of website contents that are updated regularly and credible.

### 3.3. Ability to Identify False Health Information

We randomly selected five pieces of health information that have been widely circulated on the Internet, and the respondents were asked to indicate whether they were true or false. More than 80% (80~94%) of the students were able to correctly answer four of the five questions, but only 59.2% of the students provided a correct answer regarding the item “fruits and vegetables that are darker in color are more nutritious!”, a commonly reported piece of information, indicating that a fact’s frequency of presentation and students’ ability to identify it as correct are not correlated. We found that 67.2% of adolescents applied health information acquired from the Internet in everyday life (Table 8).

### 3.4. Channels of Health Information Acquisition

Yahoo! Answers, Google, Wikipedia were the top three online sites that the adolescents used for health information, while the health websites of private institutions were used the least often, followed by the health websites of public institutions and bulletin board systems (BBSs). The participants attached more importance to and were more satisfied with health information provided by the top three online sites than with that provided by public and private health websites and BBSs. The male junior high school students’ responses indicated that they found public health websites to be unimportant channels with low satisfaction; among the other groups, public and private health websites were generally considered overemphasized, offering fair satisfaction but low importance.

### 3.5. Content of Health Information Queries by Gender

Most of the Internet health information content browsed by young people pertained to sports, fitness, and medical knowledge. The contents of health information queries varied according to the students’ age and gender. Female students were more interested in weight loss, beauty, and health than male students, while high school students more frequently searched for content in all categories than junior high school students (Table 9), indicating that the demand for health information increases with age.

The adolescents regarded most health information content as important and satisfying and were most interested in medical knowledge and sports and fitness. The junior high school students placed sexual knowledge in Quadrant 2 (overemphasized), with acceptable satisfaction but low importance, while the high school students viewed such knowledge as very important with acceptable satisfaction, placing it in Quadrant 1. Female high school students were more interested in and satisfied with beauty, health, and weight loss information than male high school students. Male and female students did not differ significantly in health awareness. The other pieces of health information mostly fell within Quadrant 1, with high satisfaction and high importance.

### 3.6. Search Channels and Health Issues

In different search engines, trending queries regarding health issues varied. Yahoo! Answers users searched for “sports and fitness” information more frequently, while Google and Wikipedia users searched for all health topics more frequently. Among Google queries, “weight loss” was the most frequent search topic, while among Wikipedia searches, “mental health” was the most frequent. BBS users browsed “sexual knowledge” most frequently. However, the information sources of BBSs are diverse, and government agencies should strengthen supervision to prevent adolescents from receiving incorrect information. For public and private health websites, queries on all health topics except for “sexual knowledge” were less frequent (Figure 1).

### 3.7. Chuchiming Index

The chuchiming index calculates the formula of (satisfaction order—importance or-der)/importance order which Prof. Dr. Chu has developed [15] and Table 10 shows that the important information of perceiving targets are Private websites (chuchiming index = 0.20) and Wikipedia (chuchiming index = 0.50) in junior girls, and private websites (chuchiming index = 0.20) and Wikipedia (chuchiming index = 0.50) in senior girls, and BBS/PTT (chuchiming index = 0.50) in senior boys. Shifting resources are BBS/PTT [16] (chuchiming index = −0.17) and Google (chuchiming index = −0.33) in junior girls, BBS/PTT (chuchiming index = −0.17) and Yahoo (chuchiming index = −0.33) in senior girls, private (chuchiming index = −0.17) and public websites (chuchiming index = −0.20) in senior boys. Balancing items are, which chuchiming index is equal to 0, public websites in girls, Google, Wikipedia and Yahoo in boys.

## 4. Discussion

Adolescents attach importance to and are satisfied with Internet content regarding most health topics, especially “medical knowledge” and “sports and fitness”. Students of different age groups and genders differ in their Internet usage and preferred health information search channels and in the content they seek. Male students are less interested in weight loss than female students [17] (Figure 1). Women have been reported to be more concerned about birth control, diet and nutrition, exercise, physical abuse, sexual abuse, and dating violence [17]. Many studies have shown that adolescents have low trust in online knowledge [18,19,20,21], which is consistent with the results of this study.

Because adolescents have fewer health problems than adults, they browse health information less frequently and spend less time interacting with it [21]. Coupled with their poor ability to identify correct Internet information [22], these behaviors reflect the urgent need to provide a quality Internet health information environment for adolescents. For example, the global COVID-19 pandemic has caused fear among the public about emerging infectious diseases, and as a result, a broad range of misinformation has spread throughout traditional media [23,24,25,26,27,28], affecting both the physical and mental health of the general public [29].

Regarding Internet misinformation, scholars believe that the mass media, healthcare organizations, community-based organizations, and other important stakeholders should establish strategic partnerships and collaboration platforms to release relevant local public health information that can help Internet users identify correct Internet health information through technologies, such as natural language learning and processing and data mining [29] and should detect and remove Internet misinformation through use ratings, comments, or expert explanations [30,31], which should be regulated and implemented through legislation [29]. For example, the Taiwanese government recently took initiative in cooperating with LINE, a social media platform, to verify text messages through artificial intelligence (AI) and machine learning [32] in an effort to solicit more good ideas.

Google, Wikipedia, and Yahoo! Answers are highly regarded search engines for health information among young people. In contrast, some dedicated health sites established by public or private institutions are rarely used and are considered unimportant health information search channels, likely because (1) people do not know which public health information sites to visit when searching for a health information topic or (2) their data query platforms are not user-friendly or do not provide results sufficiently rapidly. Most public and private health sites are inadequate in query function and health data and are not on par with Google, Wikipedia, and Yahoo! Answers. Therefore, public and private health sites should reposition themselves in the Internet health education of adolescents and strengthen their ability to identify and correct Internet misinformation.

## 5. Conclusions

For adolescents, eHealth Literacy has become a key skill in terms of health information. Given adolescents’ limited ability to recognize accurate Internet health information and the abundance of misinformation on the Internet, providing accurate Internet health information and correcting Internet misinformation are becoming increasingly important issues in online health education for adolescents.

According to the findings of this study, young people’s usage of government-created health websites is less regular, important, and satisfied than their use of websites like Qimo, Google, or Wikipedia. These platforms can be used by relevant departments to promote accurate health information. Furthermore, the management of these websites with large usage rates should be strengthened in order to prevent misinformation from being used to mislead young people.

## Figures and Tables

**Figure 1 ijerph-19-00664-f001:**
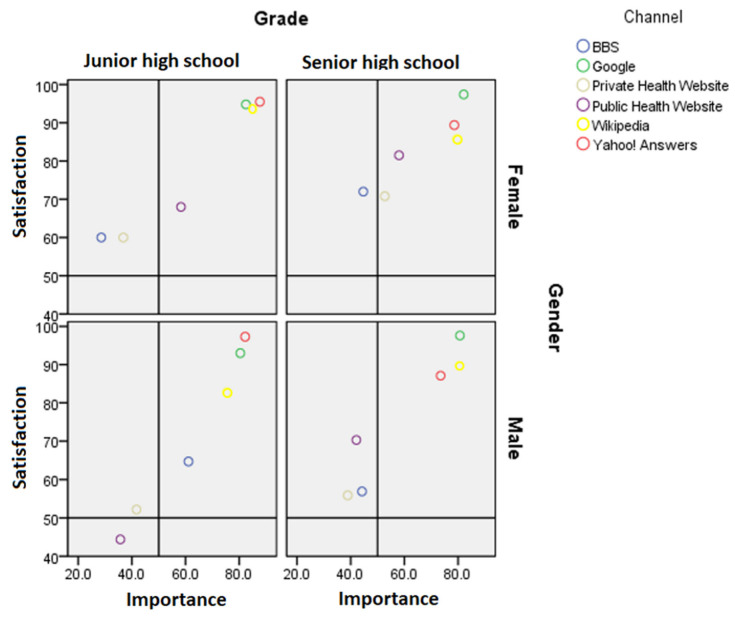
Usage of, satisfaction with, and importance of various health information search channels among students according to class (junior high school or high school) and gender.

**Table 1 ijerph-19-00664-t001:** Basic information about participant (*N* = 437).

	*n*	%
Sex		
Male	188	43%
Female	249	57%
School		
Junior high school	155	35.5%
High school	282	64.5%
Age		
13 years	36	8.2%
14 years	67	15.3%
15 years	38	8.7%
16 years	88	20.1%
17 years	123	28.1%
18 years	71	16.2%
19 years	13	3.0%
20 years	1	0.2%

**Table 2 ijerph-19-00664-t002:** The time of first Internet use and Internet use frequency and duration in the most recent month (*N* = 437).

	*n*	%
Time of first Internet use		
Before elementary school	75	17.2%
Elementary school	343	78.5%
Junior high school	17	3.9%
High school	2	0.5%
Average length of each online session		
<1 h	83	19.0%
1~2 h	171	39.1%
3~4 h	92	21.1%
>4 h	91	20.8%
Average number of days online per week		
<1 day	53	12.1%
1~3 days	215	49.2%
4~6 days	76	17.4%
Every day	92	21.1%

**Table 3 ijerph-19-00664-t003:** Frequency of Internet use in each location.

	Junior High School						*p* ^1^	High School						*p* ^1^	*p* ^1^
	*N* = 155							*N* = 282							
	Male (*N* = 84)		Female (*N* = 71)					Male (*N* = 104)		Female (*N* = 178)					
	*n*	%	*n*	%	*n*	%		*n*	%	*n*	%	*n*	%		
School	20	23.8	21	29.6	41	26.5	0.467	16	15.4	38	21.3	54	19.1	0.272	0.09
Home	80	95.2	67	94.4	147	94.8	1	100	96.2	172	96.6	272	96.8	1	0.455
Mobile internet	4	4.8	3	4.2	7	4.5	1	9	8.7	17	9.6	26	12.7	1	0.089
Friend’s house	11	13.1	5	7	16	10.3	0.292	7	6.7	9	5.1	16	5.7	0.599	0.085
library	4	4.8	4	5.6	8	5.2	1	7	6.7	10	5.6	17	6	0.797	0.831
Internet cafe	10	11.9	2	2.8	12	7.7	0.039 *	27	26	2	1.1	29	10.3	<0.001 *	0.493

^1^ χ^2^ test. * *p* < 0.05.

**Table 4 ijerph-19-00664-t004:** The purpose of Internet use.

	Junior High School			*p* ^1^	High School			*p* ^1^	*p* ^1^
	*N* = 155				*N* = 282				
	Male (*N* = 84)	Female (*N* = 71)			Male (*N* = 104)	Female (*N* = 178)			
	*n* (%)	*n* (%)			*n* (%)	*n* (%)			
entertainment	67 (79.8)	57 (80.3)	124 (80)	1	96 (92.3)	146 (82)	242 (85.8)	0.021 *	0.136
Online games	69 (82.1)	33 (46.5)	102 (65.3)	<0.001 *	69 (66.3)	32 (18)	101 (35.8)	<0.001 *	<0.001 *
Chat	51 (60.7)	47 (66.2)	98 (63.2)	0.508	86 (82.7)	135 (75.8)	221 (78.4)	0.23	0.001 *
Video download	52 (61.9)	36 (50.7)	88 (56.8)	0.194	68 (65.4)	115 (64.6)	183 (64.9)	1	0.1
Blogs	40 (47.6)	44 (62)	84 (54.2)	0.078	54 (51.9)	135 (75.8)	189 (67)	<0.001 *	0.01 *
Learning needs	32 (38.1)	43 (60.6)	75 (48.4)	0.006 *	72 (69.2)	141 (79.2)	213 (75.5)	0.064	<0.001 *
PPS	31 (36.9)	30 (42.3)	61 (45.8)	0.514	38 (36.5)	58 (32.6)	96 (34)	0.517	0.298
Find information (for non-schoolwork)	41 (48.8)	30 (42.3)	71 (45.8)	0.424	58 (55.8)	105 (59)	163 (57.8)	0.619	0.021 *
Send and receive Email	36 (42.9)	33 (46.5)	69 (44.5)	0.746	41 (39.4)	88 (49.4)	129 (45.7)	0.109	0.841
Facebook	31 (36.9)	30 (42.3)	61 (39.4)	0.514	50 (48.1)	94 (52.8)	144 (51.1)	0.461	0.021 *
Software download	36 (42.9)	11 (15.5)	47 (30.3)	<0.001 *	48 (46.2)	47 (26.4)	95 (33.7)	0.001 *	0.522
Online shopping	9 (10.7)	13 (18.3)	22 (14.2)	0.248	14 (13.5)	68 (38.2)	82 (29.1)	<0.001 *	<0.001 *
Online learning	9 (10.7)	10 (14.1)	19 (12.3)	0.625	13 (12.5)	22 (12.4)	35 (12.4)	1	1
BBS	4 (4.8)	1 (1.4)	5 (3.2)	0.376	6 (5.8)	9 (5.1)	15 (5.3)	0.789	0.473

^1^ χ^2^ test. * *p* < 0.05.

**Table 5 ijerph-19-00664-t005:** Browse contents.

	Junior High School			*p* ^1^	High School			*p* ^1^	*p* ^1^
	*N* = 155				*N* = 282				
	Male	Female			Male	Female			
	*n* (%)	*n* (%)			*n* (%)	*n* (%)			
Audiovisual entertainment	65 (77.4)	64 (90.1)	129 (83.2)	0.051	88 (84.6)	165 (92.7)	253 (89.7)	0.041 *	0.05 *
Software game	62 (73.8)	25 (35.2)	87 (43.9)	<0.001 *	71 (68.3)	43 (24.2)	114 (40.4)	<0.001 *	0.002 *
Leisure Travel	31 (36.9)	28 (39.4)	59 (38.1)	0.868	36 (34.6)	99 (56.6)	135 (47.9)	0.001 *	0.056
Life information	29 (34.5)	26 (36.6)	55 (35.5)	0.867	58 (55.8)	118 (66.3)	176 (62.4)	0.097	<0.001 *
Computer communication	28 (33.3)	22 (31)	50 (32.3)	0.863	35 (33.7)	38 (21.3)	73 (74.1)	0.025 *	0.182
Sports	31 (36.9)	10 (14.1)	41 (26.5)	0.002 *	52 (50)	31 (17.4)	83 (29.4)	<0.001 *	0.579
Arts and Humanities	13 (15.5)	26 (36.6)	39 (25.2)	0.003 *	24 (23.1)	67 (37.6)	91 (32.3)	0.012 *	0.127
Educational learning	14 (16.7)	17 (23.9)	31 (20)	0.315	30 (28.8)	52 (29.2)	82 (29.1)	1	0.04 *
Online shopping	12 (14.3)	17 (23.9)	29 (18.7)	0.15	23 (22.1)	86 (48.3)	109 (38.7)	<0.001 *	<0.001 *
Science News	16 (19)	7 (9.9)	23 (14.8)	0.119	27 (26)	17 (9.6)	44 (15.6)	0.001 *	0.89
Adult information (18 prohibited)	14 (16.7)	1 (1.4)	15 (9.7)	<0.001 *	25 (24)	3 (1.7)	28 (9.9)	<0.001 *	1
Health and medicine care	8 (9.5)	5 (7)	13 (8.4)	0.773	16 (15.4)	25 (14)	51 (14.5)	0.861	0.069
Industrial and Commercial Finance	3 (3.6)	0	3 (1.9)	0.25	11 (10.6)	11 (6.2)	22 (7.8)	0.249	0.01 *
Political and law	2 (2.4)	0	2 (1.3)	0.5	16 (15.4)	6 (3.4)	22 (7.8)	<0.001 *	0.004 *

^1^ χ^2^ test. * *p* < 0.05.

**Table 6 ijerph-19-00664-t006:** The frequency at which the students browsed health-related news or information per week.

	Junior High School		*p* *	High School		*p* *	*p* *
	*N* = 154			*N* = 282			
	Male	Female		Male	Female		
	*n* (%)	*n* (%)		*n* (%)	*n* (%)		
Days per week			0.766			0.783	0.024 *
<1 day	59 (72.8)	50 (71.4)		67 (65)	123 (69.9)		
1~3 day	18 (22.2)	18 (25.7)		32 (31.1)	48 (27.3)		
4~6 day	1 (1.2)	0		4 (3.9)	4 (2.2)		
Everyday	3 (3.7)	2 (2.9)		0	1 (0.6)		

* *p* < 0.05.

**Table 7 ijerph-19-00664-t007:** Channels for acquiring health information.

	Junior High School			*p*^1^ Value	High School			*p*^1^ Value	*p*^1^ Value
	*N* = 155				*N* = 282				
	Male	Female			Male	Female			
	*n* (%)	*n* (%)			*n* (%)	*n* (%)			
Newspapers and magazines	35 (41.7)	41 (57.7)	76 (49)	0.054	69 (66.3)	147 (82.6)	216 (76.6)	0.002 *	<0.001 *
Television news	59 (70.2)	54 (76.1)	113 (72.9)	0.471	82 (78.8)	149 (83.7)	231 (81.9)	0.338	0.037 *
Leaflets	14 (16.7)	8 (11.3)	22 (14.2)	0.366	15 (14.4)	33 (18.5)	48 (17)	0.415	0.497
Health center	19 (22.6)	19 (26.8)	38 (24.5)	0.578	22 (21.2)	29 (16.3)	51 (18.1)	0.338	0.136
Internet	51 (60.7)	57 (80.3)	108 (69.7)	0.009 *	74 (71.2)	145 (81.5)	219 (77.7)	0.054	0.084
Teachers	57 (67.9)	51 (71.8)	108 (69.7)	0.604	72 (69.2)	130 (73)	202 (71.6)	0.497	0.662
Classmates and friends	34 (40.5)	33 (46.5)	67 (43.2)	0.516	44 (42.3)	83 (46.6)	127 (45)	0.536	0.763

^1^ Chi-square test. * *p* < 0.05.

**Table 8 ijerph-19-00664-t008:** Ability to identify true or false health information (*N* = 437).

	Have Read the following Message			Agree with the following Message			
	*N*	*n*	%	*N*	*n*	%	Correct Rate
Fruits and vegetables that are darker in color are more nutritious! (O)	426	262	61.5%	407	241	59.2%	59.2%
Washing your hair every day will make you bald! (X)	425	171	40.2%	400	78	19.5%	80.5%
Drinking longan tea can treat myopia! (X)	427	67	15.7%	383	62	16.2%	83.8%
Brushing your teeth after meals will harm your dental health! (O)	425	66	15.5%	390	26	6.7%	93.3%
Removing your wisdom teeth can make your face thinner! (O)	425	57	13.4%	393	34	8.7%	91.3%

**Table 9 ijerph-19-00664-t009:** Channels for health information queries.

	Junior High School			*p*^1^ Value	High School			*p*^1^ Value	*p*^1^ Value
	*N* = 155				*N* = 282				
	Male	Female			Male	Female			
	*n* (%)	*n* (%)			*n* (%)	*n* (%)			
Yahoo! Answers	74 (90.2)	67 (94.4)	141 (92.2)	0.384	100 (97.1)	170 (95.5)	270 (96.1)	0.751	0.114
Google	53 (64.6)	52 (73.2)	105 (68.6)	0.296	77 (74.8)	146 (82.5)	223 (79.6)	0.127	0.014 *
Wikipedia	51 (62.2)	45 (63.4)	96 (62.7)	1	72 (69.2)	133 (75.6)	205 (73.2)	0.266	0.029 *
BBS	11 (14.1)	6 (8.5)	17 (11.4)	0.313	26 (25)	28 (15.9)	54 (19.3)	0.084	0.041 *
Public health website	13 (15.9)	14 (20)	27 (17.8)	0.53	10 (9.7)	29 (16.4)	39 (13.9)	0.152	0.327
Private health website	8 (9.9)	7 (9.9)	15 (9.9)	1	7 (6.8)	17 (9.6)	24 (8.6)	0.51	0.726

^1^ Chi-square test. * *p* < 0.05.

**Table 10 ijerph-19-00664-t010:** Chuchiming index.

		Junior				Senior		
		S Order	I Order	Chuchiming Index		S Order	I Order	Chuchiming Index
Girls	BBS/PTT	5	6	−0.17		5	6	−0.17
	Google	2	3	−0.33		1	1	0.00
	Private	6	5	0.20		6	5	0.20
	Public	4	4	0.00		4	4	0.00
	Wiki	3	2	0.50		3	2	0.50
	Yahoo	1	1	0.00		2	3	−0.33
Boys	BBS/PTT	4	4	0.00		6	4	0.50
	Google	2	2	0.00		1	1	0.00
	Private	5	5	0.00		5	6	−0.17
	Public	6	6	0.00		4	5	−0.20
	Wiki	3	3	0.00		2	2	0.00
	Yahoo	1	2	0.00		3	3	0.00

S: Satisfaction, I: Importance and orders are according to Figure 1. Chuchiming index > 0 indicates items need concerted improvement (perceiving targets), Chuchiming index < 0 indicates resources can be drawn from items (shifting resources), Chuchiming index = 0 indicates items can fit people’s expectation (balancing items).

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
