# Peer review of "Gender-Specific Determinants of eHealth Literacy: Results from an Adolescent Internet Behavior Survey in Taiwan"

_ijerph, 2022, doi:10.3390/ijerph19020664_

Round 1

Reviewer 1 Report

The topic is interesting  but the authors should pay attention to my in-text comments to improve their manuscript.

Author Response

Thank you very much for taking the time to write a review. Please see our response attached.

Reviewer 2 Report

Thank you for submitting your manuscript. It is an interesting topic; please allow me to suggest a few improvements or clarifications that I believe are necessary and important in ensuring the quality of your manuscript.

  1. Could you please elaborate on how the questionnaire was distributed? Is it utilising an online medium? If so, what medium did you use? Or was it physically distributed?
  2. In 2.2 Questionnaire, please indicate your ethical clearance in this subtopic as well
  3. The instrument requires further development. How many constructs are there? How many items are there? What scale is being used? Do you use, adapt, or create your own questionnaire?
  4. Please include the instrument's reliability and pilot test data.
  5. Please correct the grammatical and linguistic error

Thank you.

Author Response

(The authors gave the same response as above.)
